# Effect of Culturally Adapted Dental Visual Aids on Oral Hygiene Status during Dental Visits in Children with Autism Spectrum Disorder: A Randomized Clinical Trial

**DOI:** 10.3390/children9050666

**Published:** 2022-05-05

**Authors:** Ala Aljubour, Medhat AbdElBaki, Omar El Meligy, Basma Al Jabri, Heba Sabbagh

**Affiliations:** 1Pediatric Dentistry Department, Faculty of Dentistry, King Abdulaziz University, Jeddah 21589, Saudi Arabia; eabdulbaqi@kau.edu.sa (M.A.); omeligy@kau.edu.sa (O.E.M.); hsabbagh@kau.edu.sa (H.S.); 2Pediatric Dentistry and Dental Public Health Department, Faculty of Dentistry, Alexandria University, Alexandria 21521, Egypt; 3Pediatric Department, Faculty of Medicine, King Abdulaziz University, Jeddah 21589, Saudi Arabia; baljabri@kau.edu.sa

**Keywords:** autism spectrum disorder, dental visual aids, oral hygiene status

## Abstract

Autism spectrum disorder (ASD) prevalence has escalated in the last few decades; it is common to have children with ASD seek dental treatment. Their unique behavior patterns prevent them from attending a regular dental setting and receiving proper oral hygiene instructions (OHI); therefore, culturally adapted dental visual aids are necessary to teach them proper OHI. The aim of this study was to assess the effectiveness of culturally adapted dental visual aids in improving oral hygiene (OH) status in children with ASD. A blinded, randomized, controlled clinical trial with sixty-four children with ASD were randomly divided into two groups according to the type of dental visual aids given to them. The experimental group received culturally adapted dental visual aids developed specifically for the study, and the control group received regular dental visual aids. OH status was assessed before and after using the dental visual aids, and data were processed using SPSS version 25.0. OH status improved significantly in both groups after using the dental visual aids (*p* < 0.001, *p* < 0.001), respectively. The experimental group showed significant improvement in comparison to the control group in OH status (*p* = 0.030). Both dental visual aids showed effectiveness in improving OH status in children with ASD.

## 1. Introduction

Autism spectrum disorder (ASD) is a severe lifetime neurodevelopmental disorder characterized by significant disabilities in the individual’s social skills, poor mutual communication, in addition to having a unique pattern of rigid repetitive activities [1,2].

There is a lack of dental awareness among dentists when dealing with children with ASD and how to properly teach them oral hygiene instructions. Children with ASD exhibit different unique behavioral patterns, depending on the severity of ASD, and most of them are incapable of understanding and following instructions. Therefore, they are unable to cooperate with the dentist [3], which can make it difficult for them to be treated in a regular dental setting and receive proper oral hygiene instructions, depending on the severity of their ASD, providing an opportunity for oral problems to arise [4].

Behavior management in pediatric dentistry uses the well-known TSD, which is a very effective, yet basic technique, allowing the dentist to explain the dental procedure steps to the child to familiarize him/her with the dental instruments used in each procedure in a simple non-intimidating way [5]. However, this technique is only efficient with children who are verbally developed and can understand what the dentist is explaining. It cannot be used with children that have verbal communication deficiencies, hearing disabilities, or with children that have ASD [6].

For such patients, the most proper and efficient behavior management approach is the application of early intensive behavioral intervention to improve the performance [7]. An example of these behavioral therapies is an approach called “Treatment and Education of Autistic and Related Communication Handicapped Children” (TEACCH), which include visual aids and video modeling [8,9]. These interventions work intensively on developing language, social responsiveness, and imitation skills and improving behavior of the affected child.

Visual aids are an integral part of the TEACCH approach, comprising a structured visual guide system composed of a series of cards containing pictures, where each card can be used to explain a step in their daily routine or dental procedure [10]. It was observed that creating dental visual aids explaining tooth brushing and dental appointment steps were considered a very effective technique to teach children with ASD how to perform daily tooth brushing and gradually expose them to the steps of a dental appointment [11,12,13].

In our region, research on the effect of dental visual aids on children with ASD is limited. Sallam et al. in 2013 tested the effectiveness of three different types of dental visual aids during tooth brushing on children with ASD and concluded that the picture dental visual aids and the video modeling are useful tools in improving the oral hygiene status of children with ASD [14].

Murshid in 2017 assessed the effect of a social storybook on the behavior of children with ASD during their first dental visit and measured the tooth-brushing pattern and found that almost half of the children with ASD in their study practiced tooth brushing irregularly (57.5%) before the social story, and this percentage decreased to 35% after a five–month follow–up [15].

The aim of this study was to assess the effectiveness of the culturally adapted dental visual aids in improving oral hygiene status in children with ASD.

## 2. Materials and Methods

### 2.1. Ethical Approval

Ethical approval from the Research Ethics Committee at the Faculty of Dentistry in King Abdulaziz University (KAU) was obtained before starting the project on the 9 April 2019, under approval number 057-02-19 (Appendix A). Written consents were obtained from the parents for children’s participation in the study.

### 2.2. Clinical Trial Registration

The clinical trial was registered at www.Clinicaltrials.gov with the identification number NCT04576559. Accessed on 1 September 2019.

### 2.3. Sample Size

Sample size was calculated by using OpenEpi program, Version 3, where it showed that to have a confidence interval of 95% and a power sample of 80%, we should have a sample size of 64 children with ASD (43 males and 21 females) divided randomly into two groups, an experimental group and a control group with 32 children with ASD in each group, based on previous research in the literature [16].

### 2.4. Criteria of Participants

Children attending the ASD diagnosis clinic from January 2019 until January 2021 were all invited to participate in the study.

#### 2.4.1. Inclusion Criteria

(1) The age of the children was between 6–12 years, (2) patients with ASD diagnoses were confirmed from the patients’ files and according to the DSM–V criteria [17], (3) patients represented the typical triad symptoms of ASD: social interaction deficits, communication impairment, language impairment, and rigidity of interests, and were assessed by a professional in the field of ASD, (4) patients had no previous dental experience, (5) patients had mild to moderate ASD according to the “Childhood Autism Rating Scale” (CARS) [18].

#### 2.4.2. Exclusion Criteria

Patients with other congenital anomalies, for example cerebral palsy and Down syndrome, were excluded.

### 2.5. Study Design

The study design was a blinded, randomized, controlled clinical trial.

### 2.6. Settings and Locations

Our current study took place at the university’s dental hospital in the pediatric specialty clinics.

### 2.7. Randomization and Blinding

Children were randomly divided into two groups according to the type of dental visual aids given to them using a simple randomization technique with the main investigator tossing a coin where heads allocated the child to the experimental group and tails allocated the child to the control group. Each participant was assigned a unique serial number that corresponded with the assigned allocated group on an Excel sheet created for blindness.

The research was conducted by the main investigator (who performed all the dental procedures and measured the plaque index scores) and an observer, who recorded the plaque index scores implemented by the main investigator. Both were blinded to the patient allocation group after the first visit, where a new set of documents were used for each patient in the following dental visit that had an assigned serial number but did not mention the type of intervention given to the child. In addition, the data analyst was blinded to the data analyzed.

### 2.8. Interventions (Dental Visual Aids)

The culturally adapted dental visual aids specially created for this study to comply with our culture and society were given to the experimental group. They were created by recruiting a special artist that designed and drew colored pictures based on photographs taken of the actual setting of the pediatric dentistry clinic. They were validated by three ASD specialists, including a pediatrician that specialized in ASD, a child psychologist, and a behavior psychologist. The Arabic language was checked by an Arabic language expert before the start of this study, as shown in Figure 1 and Figure 2.

The regular dental visual aids extracted from www.ageofautism.com accessed on 9 April 2019 were given to the control group, as shown in Figure 3 and Figure 4. They were retrieved in their original English language and were explained in Arabic for the guardians and the children with ASD; a set of phrases was agreed on between the main investigator and parents to be explained daily to the children at home in their local language.

### 2.9. Methods

#### 2.9.1. Preparatory Visit

Preparatory visits were conducted at the ASD diagnosis clinic at the university’s hospital. The nature of the study was explained to the guardians by the main investigator, then there was a questionnaire that had two parts: the first part inquired about their child’s demographic data and the parents’ social status, derived from the Central Department of Statistics and Information (CDSI) [19]. The second part inquired about the child’s medical and dental history. Later they were given an appointment after one week for their child’s first dental visit Appendix B (Preparatory Visit Questionnaire).

#### 2.9.2. Pretest Survey

The demographic questionnaire in this study was pretested on random guardians to assure understanding of the questions. Accordingly, rephrasing was done to make it easily understood.

#### 2.9.3. First Dental Visit

At the pediatric dentistry clinic, the main investigator asked the child to sit in the dental chair; the only behavior management technique used with children in this visit was TSD technique.

The dental procedures in the first dental visit were all performed by a single calibrated main investigator, and another calibrated investigator (observer) charted the plaque index scoring that was measured by the main investigator.

The first assessment visit included the following for both groups: (1) oral examination, (2) plaque index scoring [20], (3) professional prophylaxis with a rotary cup and non-fluoridated prophylactic paste using a low-speed handpiece, and (4) oral hygiene instructions explained by the main investigator to the child and parents on a dental model. At the end of the appointment, each child received a copy of the dental visual aids according to their allocation group.

Guardians were instructed to read and explain these dental visual aids to their child directly for at least fifteen minutes at the same time daily [21], for a minimum period of four weeks [22,23]. Two weeks later after the first dental visit, parents of both groups were contacted to confirm that dental visual aids were being explained daily to their child.

#### 2.9.4. Second Dental Visit

Four weeks later, children in both groups were recalled. The main investigator asked the child to sit in the dental chair; the behavior management technique used in this visit was TSD.

The second assessment visit included the following for both groups: (1) oral examination, (2) plaque index scoring [20], and (3) professional prophylaxis with a rotary cup and non–fluoridated prophylactic paste using a low-speed handpiece.

All dental procedures in the second dental visit were performed by the same calibrated main investigator, and the same calibrated observer charted the plaque index scoring that was measured by the main investigator. At the end of the study, children who needed further treatment were referred to the postgraduate department at the university’s dental hospital for comprehensive dental treatment.

### 2.10. Methods of Outcomes Assessment

Assessment of the effectiveness of the culturally adapted dental visual aids in improving oral hygiene status before and after using the dental visual aids in each group and between both groups was done by measuring the plaque index scores [20], where zero indicated excellent, values between 0.1–0.9 indicated good, values between 1.1–1.9 indicated fair, and values between 2.0–3 indicated poor oral hygiene.

### 2.11. Calibration and Reliability

A calibration workshop was conducted to assure that the main investigator understood how to measure plaque index scores and to ensure that the observer fully understood how to record behavior distress signs. The intra-investigator reliability was perfect with 100%, and the intra-observer reliability was perfect with 100% agreement. The completeness of information in the assessment charts was evaluated by randomly selecting ten charts and examining them by a third clinician experienced in the field of ASD research. The inter-examiner agreement was 100%.

### 2.12. Statistical Analysis

All data were processed using the using the Statistical Package for the Social Sciences (SPSS) version 25.0 (IBM Inc., Chicago, IL, USA). Descriptive statistics frequency, percentage, and Chi–square tests were used to give an overview of the demographic and clinical characteristics of the children with ASD participating in the study. Descriptive data were described using mean and standard deviation (SD). The significance level was set at 0.05 and the level of confidence for this analysis was 95%. Wilcoxon signed–rank test was used to assess the difference before and after using dental visual aids within each group, and the Mann–Whitney test was used to compare mean ranks between both groups.

Ordinal regression analysis was calculated to assess the association between oral hygiene status after the use of the dental visual aids (dependent factor) and the type of dental visual aids, ASD severity, gender, and tooth brushing as predictors.

## 3. Results

### 3.1. Number of Participants

A total of 164 local children with ASD were invited for this study for 24 months of data collection from January 2019 through January 2021. Eighty-seven children with ASD were enrolled by their guardians using a consent form. Of the remaining seventy-seven children, sixty children declined to participate, and seventeen children met the exclusion criteria. The remaining eighty-seven children were split randomly into two groups. Forty-three children were allocated to the experimental group, thirty-two children received the allocation intervention, and the remaining eleven children did not receive the allocation intervention. Forty-four children were allocated to the control group, thirty-two children received the allocation intervention, and the remaining twelve children did not receive the allocation intervention. The sixty-four children with ASD in both groups were capable of completing the study program, as described by the “Consolidated Standards of Reporting Trials” (CONSORT) [24], as shown in Figure 5.

### 3.2. Sample Characteristics

Our current study was carried out on a group of local children with ASD aged six to twelve, the mean age being eight years and two months. In the experimental group, there were 21 males (65.62%), and 11 females (34.37%), and in the control group, there were 22 males (68.75%), and 10 females (31.25%). The distribution of gender among the groups was not statistically significant (*p* = 0.790), as shown in Table 1.

### 3.3. Children’s Characteristics

The mean age at which the children were diagnosed with ASD was 5.5 years. The severity of ASD in both groups was surveyed, and the distribution of the severity of ASD among the groups was not statistically significant (*p* = 0.209), as shown in Table 1.

The main caregiver of the child with ASD was surveyed in both groups, as shown in Table 2.

Whether the child brushed his/her teeth or not was surveyed in both groups, and the distribution of tooth brushing in children with ASD among the groups was not statistically significant (*p* = 0.578).

The frequency of tooth brushing in both groups was surveyed as well, and the distribution of frequency among the groups was not statistically significant (*p* = 0.638).

Moreover, the guardians’ assistance during tooth brushing was surveyed, and the distribution of assistance among the groups was not statistically significant (*p* = 0.083), as shown in Table 3.

### 3.4. Oral Hygiene

#### 3.4.1. Intragroup Comparison

When comparing the plaque index scores for each group in our study to assess the difference before and after the use of the dental visual aids, there was a statistically significant difference in the experimental group before and after the use of dental visual aids (*p* < 0.001), where plaque index score mean was lower after using the dental visual aids. Regarding the control group, there was also a statistically significant difference before and after using the dental visual aids where plaque index score mean was lower after the use of dental visual aids (*p* < 0.001), as shown in Table 4.

#### 3.4.2. Intergroup Comparison

We compared differences in plaque index scores before and after the use of dental visual aids between both groups. There was a statistically significant difference between both groups (*p* = 0.030), showing lower plaque index score mean rank in the experimental group in comparison to the control group (27.45, 37.55), respectively, as shown in Table 5.

The ordinal regression analysis assessed the effect of the two types of dental visual aids used on oral hygiene status of ASD children after removing the effect of other confounding factors, including child gender, ASD severity, and if the child was brushing his teeth. Culturally adapted dental visual aids was the only factor that showed statistically significant association with plaque index status and statistically improved the plaque score (AOR: 0.193, 95% CI: 0.074 to 0.507, and *p* = 0.001), as shown in Table 6.

## 4. Discussion

We compared our culturally adapted dental visual aids created especially for our study to the regular dental visual aids extracted from the Internet in improving the oral hygiene status in children with ASD.

Oral health care in children with ASD is often neglected due to difficulty accessing dental care, and their complex behavior patterns [25,26]. After reviewing the literature, our study is the first in our country to develop dental visual aids that complied with our society and represented the actual dental setting, dental staff, and dental instruments that the child encounters, whilst explaining the daily home tooth brushing steps [13]. We aimed to create a social story familiar to the children with ASD that represented their culture in their local language. This was supported by previous studies reported in the literature [27,28] where they mimicked the actual dental setting and dental staff in the child’s real life, allowing them to become familiar with the environment that they will encounter.

In comparison, children with ASD in the control group were given dental visual aids retrieved from www.ageofautism.com. Accessed on 9 April 2019. They were composed of a pictorial series that represented the steps of the dental appointment and the home tooth brushing steps in simple English phrases and were widely used by individuals with ASD around the world [29]. However, they are not commonly used by children with ASD in our region because of the language barrier. The regular dental visual aids were explained to the parents and the children with ASD in their native language to reduce possible bias that would result from comparing two dental visual aids in different languages.

The gender distribution in our study was 32.8% females and 67.2% males with a male to female ratio of 2:1. This is due to the higher prevalence of ASD in males, which coincides with previous studies reported in the literature [30,31,32].

The present study was conducted on a group of local children with ASD aged from six to twelve. The selection of this age group started at six years old because children with ASD at this age can somehow have a good grasp on a toothbrush for tooth brushing [14] and ended at 12 years to eliminate the chance of any puberty hormones to negatively interfere with their behavior patterns [33,34].

The children were all selected according to CARS [18]. We included children with mild to moderate levels of ASD. This selection was based on the ability of children with mild ASD to speak and make simple conversation, while children with moderate ASD spoke with single words and were not able to make a conversation but complied with short simple phrases, such as orders given to them during the dental appointment. This selection was supported by previous studies in the literature [28,35].

Guardians in both groups in our study were instructed to explain the dental visual aids to their children for at least fifteen minutes daily and a minimum period of four weeks, as this is the minimum period required to modify any behavior in children with ASD effectively and prepare them to learn a new skill [21,22,23].

Most of the guardians in our current study reported that they brushed their children’s teeth daily (71.9%), while the rest of them reported that they do not brush their children’s teeth at all (28.1%). Caregivers who did not brush their children’s teeth reported that their child refused to brush and behaved negatively when attempting brushing. These results came in agreement with the studies in the literature [14,34]; this might be due to the uncooperative behavioral pattern in children with ASD during tooth brushing.

In the present study, 15.6% of the children brushed alone, while 57.8% of the caregivers reported they helped their children during brushing; a study by Murshid in 2017 confirmed these findings and reported that caregivers of children with ASD assist their children during brushing. This might be due to the physical impairments and poor manual skills, which are counted amongst the characteristics of ASD and which make physical assistance necessary during daily tooth brushing [4,15,36,37].

Based on our findings, caregivers play a major role in their child’s life when taking care of their oral health; thus, it is crucial to educate them about proper oral hygiene measures, the appropriate techniques of toothbrushing, and modified toothbrushes that can greatly aid in achieving optimum oral health [38].

In our study, we applied non-fluoridated prophylactic paste because parents of children with ASD were concerned about the fluoride component in the paste and argued that it might worsen the case of ASD. However, they were assured that the prophylactic paste did not contain any fluoride, and we supported our argument with evidence from the literature that fluoride does not increase the severity of ASD [39].

In the present study, we measured plaque index scores before and after applying both types of dental visual aids using the Plaque Index Scoring System [20]. The results of oral hygiene status of children with ASD were measured at baseline in the first dental visit for both groups combined and showed that none of the children had excellent oral hygiene. Only 18% had good oral hygiene, 50% of the children had fair oral hygiene, and 28.1% had poor oral hygiene. These results were reported, although 71.9% of the guardians brushed their children’s teeth daily. These findings were consistent with results obtained from a study by Magoo et al. in 2015 in which most children had poor oral hygiene, although 86.5% of the guardians reported they assisted their child with brushing [38]. This might be due to irregular brushing frequency, improper brushing technique, or the lack of cooperation in children with ASD during brushing.

There was a statistically significant effect on the plaque index scores in children with ASD before and after using both types of dental visual aids in both groups. These findings are in agreement with a study by Sallam et al. in 2013 that assessed the oral hygiene status. They tested three different types of dental visual aids on children with ASD and observed significant improvement in oral hygiene status in the picture and video dental visual aid groups, while there was no significant difference observed in oral hygiene status in the third group that received modeling by using a plastic cast of the upper and lower jaws [14]. The improvement in oral hygiene status observed in these two groups might be explained by the positive effect of dental visual aids on the learning process in children with ASD and due to the nature of children with ASD where they respond better with visual communication using pictures and video rather than communication with verbal directions, such as TSD.

Furthermore, Murshid in 2017, reported that improved oral hygiene status was observed in children with ASD that received dental visual aids composed of a social story that prepared them for their first dental visit. Those children showed a significant difference in tooth-brushing patterns. The parents in their study reported that 57.5% of the children practiced irregular tooth-brushing habits when they first received the social story; this percentage was reduced to 35% after using the social story [15]. These findings are in agreement with the results found in our present study where improved oral hygiene status was observed in both groups; this might be due to the repeated daily reading of the social story by the parents before coming to the dental appointment, which helped the children with ASD become familiar with the upcoming dental appointment.

## 5. Conclusions

Both culturally adapted dental visual aids and regular dental visual aids were effective in improving the oral hygiene status in children with ASD.

## 6. Recommendations

Early referral to a pediatric dentist once a child is diagnosed with ASD is essential to promote good oral health.

Future research is suggested to test the effectiveness of a digital copy of the culturally adapted dental visual aids in comparison to the hardcopy.

## 7. Limitations

We included only children with mild to moderate ASD.

## 8. Implications for Practice

Due to the effectiveness of this protocol, it could be considered as future guidelines for teaching children with ASD proper oral hygiene instructions.

## Figures and Tables

**Figure 1 children-09-00666-f001:**
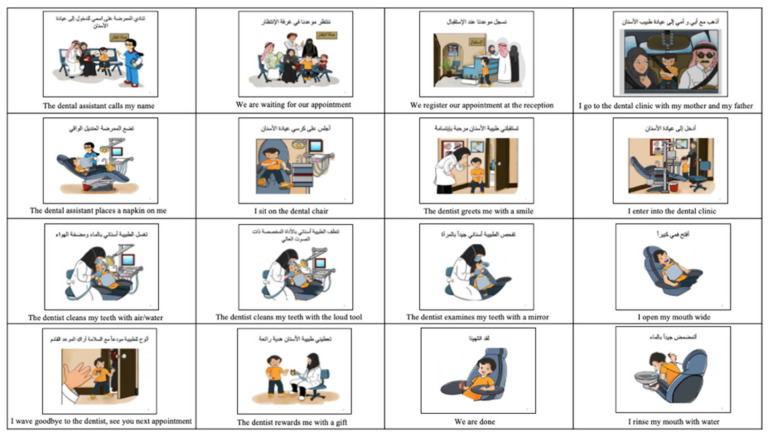
The culturally adapted dental visual aids: visiting the dentist.

**Figure 2 children-09-00666-f002:**
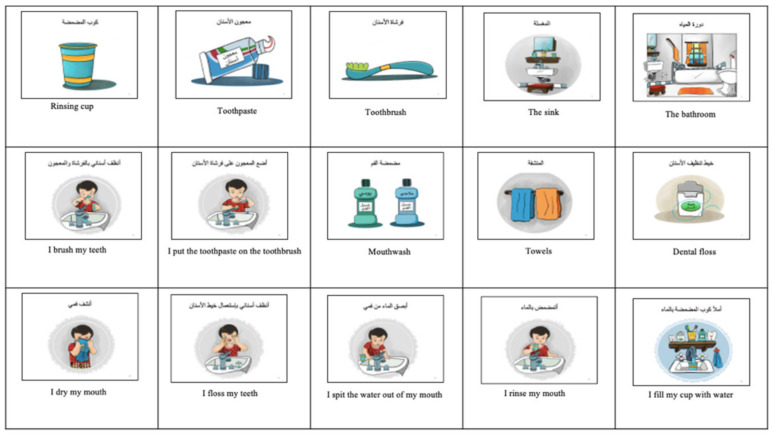
The culturally adapted dental visual aids: brushing teeth at home.

**Figure 3 children-09-00666-f003:**
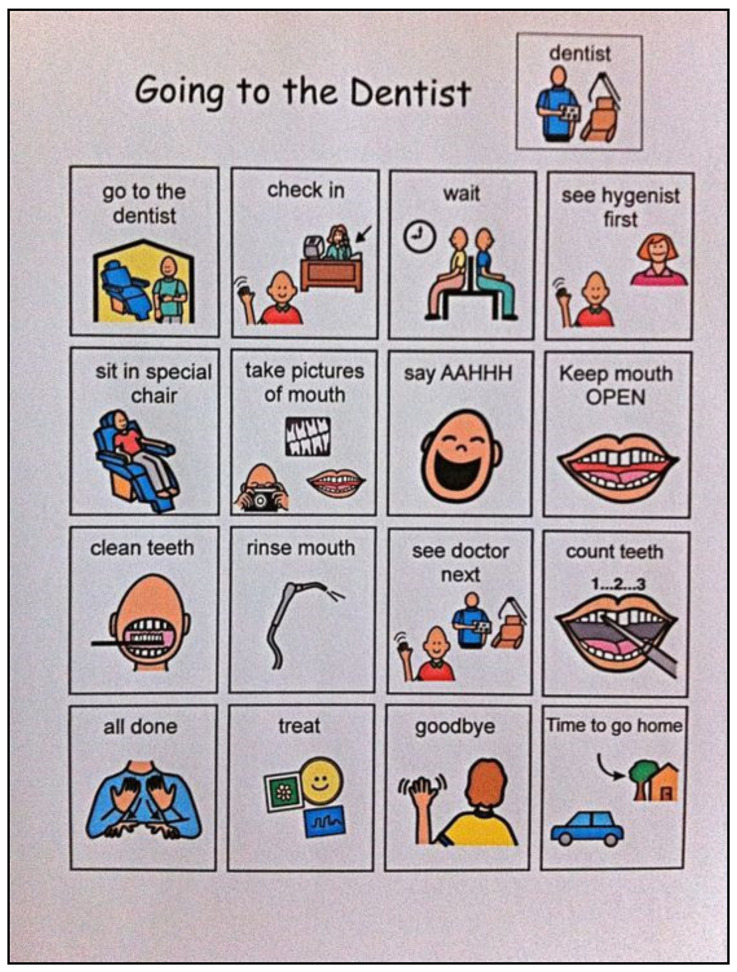
The regular dental visual aids: going to the dentist.

**Figure 4 children-09-00666-f004:**
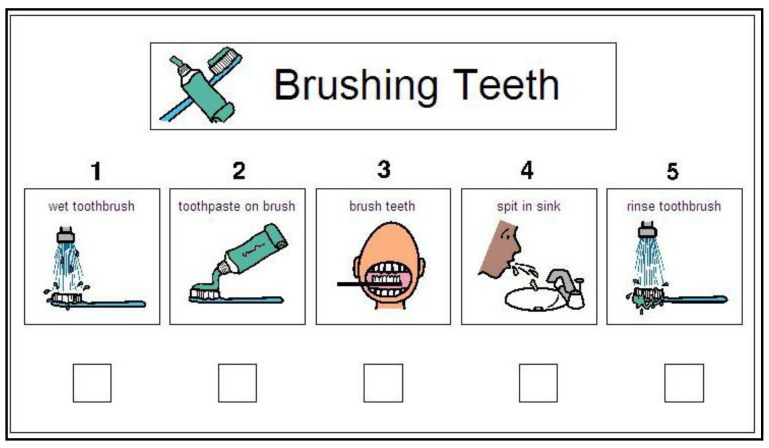
The regular dental visual aids: brushing teeth.

**Figure 5 children-09-00666-f005:**
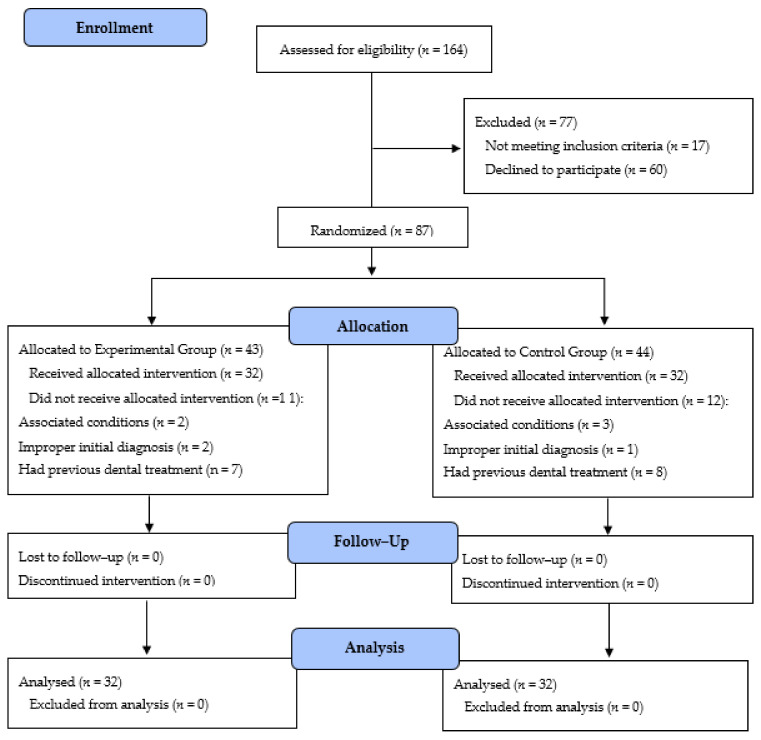
The CONSORT transparent reporting of trials. *n*: number of children with ASD.

**Table 1 children-09-00666-t001:** The Distribution of Subjects’ Gender.

The Distribution of Subjects’ Gender					
Gender	Experimental Group		Control Group		*p*-Value
	*n*	%	*n*	%	
**Male**	21	65.62	22	68.75	0.790
**Female**	11	34.37	10	31.25	
**Total**	32	100	32	100	
**The Distribution of ASD Severity**					
	**Experimental Group**		**Control Group**		***p*-Value**
**Severity of ASD**	** *n* **	**%**	** *n* **	**%**	0.209
**Mild**	17	53.12	12	37.50	
**Moderate**	15	46.87	20	62.50	
**Total**	32	100	32	100	

*n*: number of children with ASD. Pearson Chi–square test.

**Table 2 children-09-00666-t002:** Distribution of Caregivers Among Children with ASD.

	Experimental Group		Control Group	
Main Caregiver	*n*	%	*n*	%
**Mother**	25	78.12	23	71.87
**Father**	1	3.12	1	3.12
**Nurse**	0	0.0	1	3.12
**Housemaid**	6	18.75	7	21.87
**Total**	32	100	32	100

*n*: number of children with ASD.

**Table 3 children-09-00666-t003:** Distribution of Tooth Brushing, Frequency, and Caregivers’ Assistance Among Children with ASD.

	Experimental Group		Control Group		*p*-Value
**Tooth Brushing**	** *n* **	**%**	** *n* **	**%**	0.578
**Yes**	22	68.75	24	75	
**No**	10	31.25	8	25	
**Total**	32	100	32	100	
**Brushing Frequency**					
**1/day**	13	40.62	13	40.62	0.638
**2/day**	8	25	11	34.37	
**3 and more/day**	1	3.12	0	0.0	
**NA**	10	31.25	8	25	
**Total**	32	100	32	100	
**Caregivers’ Assistance**					
**Yes**	15	46.87	22	68.75	0.115
**No**	7	21.87	2	6.25	
**NA**	10	31.25	8	25	
**Total**	32	100	32	100	

*n*: number of children with ASD. NA: non–applicable; Pearson chi–square test.

**Table 4 children-09-00666-t004:** Comparison Between Oral Hygiene Status Before and After the use of the Dental Visual Aids Within each Group.

	Experimental Group						Control Group					
	Before			After			Before			After		
	Mean	SD	MED	Mean	SD	MED	Mean	SD	MED	Mean	SD	MED
**Plaque Index Score**	1.445	0.567	1.4125	0.45	0.501	0.25	1.724	0.696	1.8750	1.10	0.684	1.06
***p*-value**	<0.001 *						<0.001 *					

MED: median; SD: standard deviation; Wilcoxon signed–rank test; *: statistically significant *p* < 0.05.

**Table 5 children-09-00666-t005:** Comparison Between Differences in Oral Hygiene Status Before and After the Use of the Dental Visual Aids Between Groups.

	Experimental Group		Control Group		*p*-Value
	Mean Rank	Sum of Ranks	Mean Rank	Sum of Ranks	
**Plaque Index Score**	27.45	878.50	37.55	1201.50	0.030 *

*: statistically significant *p* < 0.05. Mann–Whitney test.

**Table 6 children-09-00666-t006:** Ordinal regression analysis assessing the association between oral hygiene status after the use of the dental visual aids and the type of dental visual aids, ASD severity, gender, and tooth brushing as predictors.

Variables		*p* Value	AOR	95% Confidence Interval
**ASD severity**	**Mild**	0.160	0.521	0.210–1.293
	**Moderate**	.	1	.
**Gender**	**Male**	0.842	1.106	0.409–2.996
	**Female**	.	1	.
**Brushing**	**Yes**	0.458	0.699	0.271–1.803
	**No**	.	1	.
**Dental Visual Aids**	**Experimental Group**	0.001 *	0.193	0.074–0.507
	**Control Group**	.	1	.

*: statistically significant *p <* 0.05. AOR: Adjusted Odds Ratio.

## Data Availability

The data supporting the conclusions of this study can be obtained upon request to the corresponding author at aaljubour@stu.kau.edu.sa.

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
