# Peer review of "Effect of Culturally Adapted Dental Visual Aids on Oral Hygiene Status during Dental Visits in Children with Autism Spectrum Disorder: A Randomized Clinical Trial"

_children, 2022, doi:10.3390/children9050666_

Round 1

Reviewer 1 Report

Dear Authors,

The paper deals with a very relevant topic. The oral health of individuals with ASD and the approaches to prevention and treatment deserve careful attention. In my opinion the article presents a valuable experience for future readers. I have the following few comments and suggestions:

  • Line 107 Figure 2. check if figure 2 is correct. Above are unrelated Arabic words in the figures. I suggest to add the English translation
  • Line 381-382  increase the conclusion (add study limitations)
  • Line 404  I suggest updating older bibliographic references, such as Kanner (1943)
  • Line 433 I suggest updating older bibliographic references  such as Silness (1964)

Author Response

Dear Editor in Chief:

With this letter, I am sending you our revised manuscript entitled, “Effect of Culturally Adapted Dental Visual Aids on Oral Hygiene Status During Dental Visits in Children with Autism Spectrum Disorder: A Randomized Clinical Trial”.

The reviewers made many helpful comments that we incorporated into our attached revised document. We are grateful for their careful review because we believe the manuscript is now superior to our original.

Reviewer 1

Dear Authors,

The paper deals with a very relevant topic. The oral health of individuals with ASD and the approaches to prevention and treatment deserve careful attention. In my opinion the article presents a valuable experience for future readers. I have the following few comments and suggestions:

  • Line 107 Figure 2. check if figure 2 is correct. Above are unrelated Arabic words in the figures. I suggest to add the English translation:

English translation has been added to the figures.

  • Line 381–382 increase the conclusion (add study limitations):

Conclusions (Line 504) and Limitations (Line 512) were revised and modified.

  • Line 404 I suggest updating older bibliographic references, such as Kanner (1943): An updated reference has been added as follows:
  • Harris, Leo Kanner and autism: a 75–year perspective. Int. Rev. Psychiatry. 2018, 30, 3–17. [PubMed]

  • Line 433 I suggest updating older bibliographic references such as Silness (1964):

An updated reference has been added as follows:

  • World Health Organization (WHO). Oral Health Surveys Basic Methods,by World Health Organization Fifth Edition; 2013.

I hope that you find our revisions to the manuscript and our responses here satisfactory. If you have further questions or concerns, please contact me.

Sincerely

Ala Aljubour

Reviewer 2 Report

Dear authors, the submitted study is quite interesting. I can suggest some improvements :

  • lines 17-22, ASD is a pathology that can have different degrees, not all children with ASD may have difficulties in dental setting, please rephrase
  • line 25 TEACCH is not a new technique but is well documented from several years in literature, as you references suggests
  • lines 49-53 are in a different style, please correct
  • please pre-test survey as supplementary file and the correct citation in the paper
  • please clarify the TSD technique cited and explain this technique in the introduction section
  • some typos are present in the text, please correct
  • paragraph 3.3: results reported in tables should not be repeated in the text
  • line 291, some reviews are published on this subject, please add an appropriate reference
  • line 305 "to reduce possible bias" could be more appropriate, please reformulate 
  • the difference in language between the two aids should be indicated among the limitations of the study and discussed, along with the difference between the two groups in the verbal explanation of the aid in English. please add this explanation in the discussion section
  • paragraph 4 includes redundant sentences and repetition of sentences from introduction and results section, please reformulate
  • Among the limitations of the study should be mentioned the lack of analysis of the first support provided, regarding the part of the visit, on the collaboration of children during the visit
  • if it is not possible to perform this type of analysis, among the limitations of the study, should be mentioned the lack of analysis of the parameters assessed by the questionnaire and the improvement of oral hygiene, such as the number of toothbrushing per day between groups

Author Response

Dear Editor in Chief:

With this letter, I am sending you our revised manuscript entitled, “Effect of Culturally Adapted Dental Visual Aids on Oral Hygiene Status During Dental Visits in Children with Autism Spectrum Disorder: A Randomized Clinical Trial”.

The reviewers made many helpful comments that we incorporated into our attached revised document. We are grateful for their careful review because we believe the manuscript is now superior to our original.

Reviewer 2

Dear authors, the submitted study is quite interesting. I can suggest some improvements :

  • lines 17–22, ASD is a pathology that can have different degrees, not all children with ASD may have difficulties in dental setting, please rephrase:

Rephrasing was done as follows (lines 35–40).

“Children with ASD exhibit different unique behavioral patterns, according to the severity of ASD, most of them are incapable of understanding and following instructions. Therefore, they are unable to cooperate with the dentist [3], which can make it hard for them to be treated in a regular dental setting and receive proper oral hygiene instructions depending on their severity of ASD, giving a chance for oral problems to arise [4].”

  • line 25 TEACCH is not a new technique but is well documented from several years in literature, as you references suggests:

The word “new” was deleted.

  • lines 49–53 are in a different style, please correct:

Format of lines 4953 (now lines 8588) was checked and corrected.

  • please pre–test survey as supplementary file and the correct citation in the paper:

The pre–test survey was added as a supplementary file and was cited as Appendix II (Preparatory Visit Questionnaire).

  • please clarify the TSD technique cited and explain this technique in the introduction section:

An explanation of the TSD technique has been added to the introduction section in lines 41–56.

  • some typos are present in the text, please correct:

Spelling was checked and corrected.

  • paragraph 3.3: results reported in tables should not be repeated in the text:

Paragraph 3.3 was revised, and repetitions were deleted.

  • line 291, some reviews are published on this subject, please add an appropriate reference:

An appropriate reference has been added as follows:

Aljubour, A.; AbdElBaki, M.; El Meligy, O.; Al Jabri, B.; Sabbagh, H. Effectiveness of dental visual aids in behavior management of children with autism spectrum disorder: a systematic review. J. Child Health Care. 2021, 50, 83–107. [CrossRef]

  • line 305 "to reduce possible bias" could be more appropriate, please reformulate:

The phrase has been reformulated.

  • the difference in language between the two aids should be indicated among the limitations of the study and discussed, along with the difference between the two groups in the verbal explanation of the aid in English. please add this explanation in the discussion section:

The difference in language between the two aids has been mentioned in the methodology, lines 149–152 and in discussion, lines 426–428.

  • paragraph 4 includes redundant sentences and repetition of sentences from introduction and results section, please reformulate:

Paragraph 4 has been reformulated and repetitions were deleted.

  • Among the limitations of the study should be mentioned the lack of analysis of the first support provided, regarding the part of the visit, on the collaboration of children during the visit:

Thank you for this suggestion. To assess the child collaboration and its effect on the outcome, we conducted a regression analysis that assist the association between the main study outcome (Oral Hygiene Status) and the different variable related to child collaboration including type of Dental Visual Aids, ASD severity, and tooth brushing. We also included child’s gender as an additional predictor (Table 6).

  • if it is not possible to perform this type of analysis, among the limitations of the study, should be mentioned the lack of analysis of the parameters assessed by the questionnaire and the improvement of oral hygiene, such as the number of toothbrushing per day between groups:

Accordingly, we conducted a regression analysis (Table 6) and inserted its proper citation.

I hope that you find our revisions to the manuscript and our responses here satisfactory. If you have further questions or concerns, please contact me.

Sincerely

Ala Aljubour
